# The Effect of *Limosilactobacillus fermentum* MG4717 on Oral Health and Biosafety

**DOI:** 10.3390/microorganisms13071600

**Published:** 2025-07-07

**Authors:** Jeong-Yong Park, Ji Yeon Lee, YongGyeong Kim, Byoung-Kook Kim, Soo-Im Choi

**Affiliations:** MEDIOGEN, Co., Ltd., Biovalley 1-ro, Jecheon-si 27159, Republic of Korea; pjy@mediogen.co.kr (J.-Y.P.); ljy@mediogen.co.kr (J.Y.L.); kyk@mediogen.co.kr (Y.K.); kbk@mediogen.co.kr (B.-K.K.)

**Keywords:** *Limosilactobacillus fermentum* MG4717, periodontopathogen, *mgl*, whole-genome sequencing

## Abstract

Oral diseases such as periodontitis and dental caries, as well as conditions related to oral health such as halitosis, are closely associated with dysbiosis of the oral microbiota and continue to pose significant public health challenges worldwide. With the increasing resistance to existing antibiotics and side effects of chemical disinfectants, probiotics have emerged as promising alternatives for oral healthcare. This study aimed to evaluate the oral health efficacy and probiotic properties of *Limosilactobacillus fermentum* (*L. fermentum*) MG4717 isolated from the human oral cavity. *L. fermentum* MG4717 showed notable antimicrobial activity against the key oral pathogens *Streptococcus mutans* (*S. mutans*), *Aggregatibacter actinomycetemcomitans* (*A. actinomycetemcomitans*), *Porphyromonas gingivalis* (*P. gingivalis*), and *Fusobacterium nucleatum* (*F. nucleatum*) and effectively inhibited biofilm formation. Additionally, *L. fermentum* MG4717 significantly downregulated methionine gamma-lyase *(mgl*) mRNA expression in *P. gingivalis*, which is implicated in halitosis and pathogenicity. *L. fermentum* MG4717 strongly adhered to the KB and HT-29 epithelial cells and exhibited good resilience under simulated gastrointestinal conditions. Whole-genome sequencing (WGS) and average nucleotide identity (ANI) analysis confirmed strain identity (98.73% average nucleotide identity with *L. fermentum* DSM20052) and the absence of transferable antibiotic resistance genes. Safety assessments revealed no cytotoxicity, hemolytic activity, or bile salt hydrolase activity. These findings suggest that *L. fermentum MG4717* has the potential to be used as a safe and effective oral probiotic beneficial for oral health.

## 1. Introduction

Oral diseases, which are highly prevalent chronic conditions worldwide, primarily include caries, dental caries, and periodontal disease, while related conditions such as halitosis are commonly associated with these diseases [1]. Approximately 3.5 billion people worldwide are affected by oral diseases, with the associated social and economic burden continuing to rise, according to a 2015 report by the Institute for Health Metrics and Evaluation on the Global Burden of Disease [2].

Periodontitis is an inflammatory disease of the tooth-surrounding tissues, i.e., the gingiva and alveolar bone. It is initiated by periodontitis-associated bacteria that colonize the dental calculus on the tooth surface and subsequently proliferate, leading to inflammation of the supporting periodontal structures [3]. Periodontitis is primarily initiated and caused by *Porphyromonas gingivalis* (*P. gingivalis*) and *Fusobacterium nucleatum* (*F. nucleatum*), which are classified as orange and red complexes [4]. Additionally, *Aggregatibacter actinomycetemcomitans* (*A. actinomycetemcomitans*), a facultative anaerobic, Gram-negative, rod-shaped member of the indigenous human oral microbiota, is a major pathogen involved in the pathogenesis of periodontal disease, particularly contributing to the rapid and severe periodontal breakdown characteristics of aggressive periodontitis [5].

Halitosis is defined as an unpleasant odor originating from the mouth, caused by L-cysteine and L-methionine produced by food intake [6]. *P. gingivalis*, a representative Gram-negative anaerobic bacterium, is a major pathogen that produces volatile sulfur compounds (VSCs) such as hydrogen sulfide and methyl mercaptan, thereby contributing to halitosis [7,8]. *Streptococcus mutans* (*S. mutans*) is a primary cause of dental caries. *S. mutans* initiates the demineralization of tooth enamel by producing organic acids through carbohydrate fermentation, which lowers the pH of the oral environment and causes caries [9]. *S. mutans* forms an initial oral biofilm community by interacting with other pathogens via specific adhesion receptor mechanisms [10]. This environment may promote the growth and metabolic activity of VSC-producing bacteria, indirectly contributing to halitosis [11]. Moreover, the acidic microenvironment created by *S. mutans* can lead to dysbiosis, favoring the proliferation of anaerobic bacteria implicated in malodor production [12].

Many studies have been conducted to suppress harmful oral bacteria using antibiotics such as penicillin and tetracycline to maintain oral health. However, clinical studies have been limited, leading to increased antibiotic resistance [13]. Additionally, oral disinfectants such as chlorhexidine are often used to treat patients with periodontitis or alleviate bad breath by attaching to tooth enamel and suppressing oral microorganisms; however, their use can disrupt the oral microbiota by targeting both harmful and beneficial bacterial populations [14]. Oral probiotics are defined as “live microorganisms that, when administered in adequate amounts, confer health benefits on the host,” specifically within the oral cavity [15]. Unlike conventional antibiotics, probiotics maintain or restore a healthy microbial balance by inhibiting pathogenic bacteria, modulating immune responses, and enhancing oral microbiome integrity [16]. In the oral cavity, these beneficial bacteria can compete with harmful species for adhesion sites, produce antimicrobial compounds such as bacteriocins, and interfere with biofilm formation [17].

Commonly studied oral probiotics include *Streptococcus salivarius*, *Limosilactobacillus reuteri*, and *Limosilactobacillus fermentum* (*L. fermentum*), which have demonstrated inhibitory effects against *S. mutans* and *P. gingivalis,* the major pathogens involved in dental caries, periodontitis, and halitosis, respectively [18,19]. Therefore, this study aimed to evaluate the antimicrobial activity of *L. fermentum* MG4717, isolated from the human oral cavity, against representative oral pathogens, including *P. gingivalis*, *A. actinomycetemcomitans*, and *S. mutans*.

## 2. Materials and Methods

### 2.1. Bacterial Strains and Culture Conditions

*L. fermentum* strains MG4681, MG4684, MG4697, MG4700, MG4712, MG4717, and MG4737 were isolated from healthy human oral cavities (Table 1). For the in vitro study, *L. fermentum* was inoculated in De Man, Rogosa, and Sharpe (MRS; Difco, Detroit, MI, USA) broth and cultured at 37 °C for 24 h. All strains were registered on the National Center for Biotechnology Information database using the Basic Local Alignment Search Tool.

The oral pathogens *S. mutans* KCTC3065, *A. actinomycetemcomitans* KCTC2581, *P. gingivalis* KCTC5352, *F. nucleatum* ssp. *nucleatum* KCTC2640, and *F. nucleatum* ssp. *animalis* KCTC15571 were obtained from the Korean Collection for Type Cultures (Daejeon, Republic of Korea). Both *S. mutans* and *A. actinomycetemcomitans* were cultured in brain heart infusion (BHI) agar (Difco) at 37 °C for 48 h. After incubation, a single colony was transferred to BHI broth. *P. gingivalis* was cultured in tryptic soy agar (BD Bioscience, Franklin Lakes, NJ, USA) supplemented with 5 µg/mL hemin, 1 µg/mL vitamin K_1_, and 5% sheep blood on a plate at 37 °C for seven days. Then, the colonies were subcultured into half-BHI broth supplemented with 5 µg/mL yeast extract, 5 µg/mL hemin, and 1 µg/mL vitamin K_1_. All oral pathogens were incubated under anaerobic conditions.

### 2.2. Preparation of Cell-Free Supernatant (CFS) and Whole-Cell Lysate (WC)

For the in vitro study, the CFS was prepared as follows. To prepare CFS, the *L. fermentum* strains were adjusted to an OD_600_ of 1.0 (1 × 10^8^ CFU/mL) and subcultured at 37 °C for 18 h. Then, the strains were centrifuged at 4000× *g* for 15 min at 4 °C, and the supernatants were adjusted to pH 7.4 and filtered through a 0.22 μm polytetrafluoroethylene membrane filter (ADVANTEC, Tokyo, Japan).

For the preparation of the WC used in cytotoxicity assays on HT-29 cells, *L. fermentum* MG4717 was subcultured at 37 °C for 18 h. Then, *L. fermentum* MG4717 was centrifuged at 4000× *g* for 15 min at 4 °C, and the pellets were washed three times with sterile phosphate-buffered saline (PBS, pH 7.4). The washed pellets were resuspended in PBS, adjusted to a 1 × 10^8^ CFU/mL, and disrupted through ultrasonication using a sonicator (KOR PROTECH CO., Ltd., Seoul, Republic of Korea). The prepared samples were stored at −80 °C until use.

### 2.3. Inhibitory Effect of L. fermentum Strains Against Oral Pathogens

To investigate the inhibitory effects of *L. fermentum* strains on *S. mutans*, *A. actinomycetemcomitans*, and *P. gingivalis*, strains were prepared as previously described [1]. The oral pathogens were adjusted to a concentration of 1 × 10^8^ CFU/mL. Then, 180 µL samples of *S. mutans, A. actinomycetemcomitans*, and *P. gingivalis* were inoculated onto a 96-well plate (2 × 10^5^, 2 × 10^6^, and 2 × 10^6^ CFU/well, respectively) with 20 µL of 10% CFS of *L. fermentum*. Each plate was incubated in an anaerobic incubator for 24 h, 48 h, and 4 days, respectively. The inhibitory effects of *L. fermentum* strains on oral pathogens were measured at 600 nm using a microplate reader (BioTek, Winooski, VT, USA).

### 2.4. Biofilm Formation

To evaluate the inhibitory effects of *L. fermentum* strains on biofilm formation by oral pathogens, a crystal violet assay was performed with a few modifications [20]. Oral pathogens were diluted to a concentration of 1 × 10^8^ CFU/mL.

*S. mutans* (1 × 10^4^ CFU/well) was inoculated onto a 96-well plate under anaerobic conditions for 12 h to allow initial biofilm formation. Then, the plates were cultured for an additional 24 h with 10% CFS from each strain.

*A. actinomycetemcomitans* (1 × 10^7^ CFU/well) was inoculated onto a 96-well plate under anaerobic conditions for 24 h. Then, the plates were cultured for an additional 24 h with 10% CFS from each strain.

*P. gingivalis* was inoculated onto a 96-well plate at a concentration of 2 × 10^6^ CFU/well and incubated for five days to allow initial biofilm formation. Then, the plates were cultured for an additional 24 h with 10% CFS from each strain.

*F. nucleatum* ssp. *nucleatum* and *F. nucleatum* ssp. *animalis* were inoculated onto a 96-well plate at a concentration of 1 × 10^7^ CFU/well and incubated for 24 h to allow initial biofilm formation. Then, the plates were cultured for an additional 24 h with 10% CFS from each strain.

After incubation, the plates were washed thrice with distilled water and dried. Then, the oral pathogens were stained with 0.1% crystal violet for 15 min, washed three times with distilled water, dried, and dissolved in 95% ethanol. The absorbance of each well was measured at 575 nm using a microplate reader (BioTek).

### 2.5. Disk Diffusion Assay

A disk diffusion assay was performed to evaluate the antibacterial activity of *L. fermentum* MG4717 against *P. gingivalis*. *P. gingivalis* (BAA308; American Type Culture Collection, Manassas, VA, USA) was streaked onto ATCC2722 agar. A single colony was subsequently cultured anaerobically in ATCC 2722 broth, and the final concentration was adjusted to 1 × 10^3^ CFU/mL. The *P. gingivalis* suspension was spread uniformly on ATCC 2722 agar plates. Sterile paper disks (8 mm, MicroAnalytix, Auckland, New Zealand) were loaded with 40 μL of *L. fermentum* MG4717 culture medium or culture supernatant. Additionally, 4 μL aliquots of each sample were directly applied onto the agar surface. The plates were then incubated anaerobically at 37 °C for 24 h. Antibacterial activity was determined by measuring the diameter of the inhibition zones formed around the disks, with the appearance of a clear zone indicating an inhibitory effect against *P. gingivalis*.

### 2.6. Analysis of Methionine Gamma-Lyase (mgl) mRNA Expressions

To investigate whether the *mgl* mRNA expression of *P. gingivalis* was repressed by *L. fermentum* MG4717, *P. gingivalis* was cultured in the presence of *L. fermentum* MG4717. Total RNA was isolated using NucleoZol (Macherey-Nagel, Düren, Germany) according to the manufacturer’s instructions. cDNA was prepared from the isolated mRNA using the Maxime RT PreMix (iNtRON, Seongnam-si, Republic of Korea). mRNA expression was analyzed using the CFX96 system (Bio-Rad, Hercules, CA, USA) with IQ SYBR Green Supermix (Bio-Rad) and the following primers: *mgl* (forward): 5′-TCGTGCTTATGAGCGATGTC-3′, *mgl* (reverse): 5′-GGAAGTCACCCTCGTGGATA-3′, and *P. gingivalis*-specific 16S rRNA (forward): TACCCATCGTCGCCTTGGT, *P. gingivalis*-specific 16S rRNA (reverse): CGGACTAAAACCGCATACACTTG. Relative quantitative expression was analyzed using the 2^−ΔΔCT^ method. The mRNA levels of *mgl* were normalized via amplification of the 16S rRNA of *P. gingivalis* as an internal control.

### 2.7. Cell Culture

Human oral epithelial KB cells (KCLB, Seoul, Republic of Korea) and human colon adenocarcinoma HT-29 cells (KCLB) were cultured in Dulbecco’s modified Eagle medium (Gibco, Grand Island, NY, USA) supplemented with 10% fetal bovine serum (Gibco) and 1% penicillin-streptomycin (Gibco). The cells were subcultured to 70–80% confluency.

### 2.8. Adhesion Ability on KB Cells

KB cells were seeded in 24-well plates at a density of 2 × 10^5^ cells/well and cultured until a monolayer formed. *L. fermentum* MG4717 (1 × 10 CFU/mL) was added to each well, and the cells were incubated for 2 h. After incubation, the cells were washed three times with PBS (pH 7.4) to remove non-adherent bacterial cells. KB cells were lysed with PBS, and the adhesion rate (%) was determined based on colony counts on MRS agar plates and subsequently calculated using the following formula: Adhesion rate (%) = log (adherent counts, CFU/mL)/log (initial counts) × 100.

To determine the adherence of bacterial cells per host cell, the number of adhered bacterial cells per epithelial cell was calculated using the following formula: No. of adhered bacteria per epithelial cell = (adhered counts, CFU/mL)/(initial epithelial cell counts, cells/mL).

### 2.9. Whole-Genome Sequencing (WGS)

WGS was performed as previously described [1]. Genomic DNA of *L. fermentum* MG4717 was extracted using the PureLink™ Microbiome DNA purification kit (Invitrogen, MA, USA) following the manufacturer’s instructions. WGS was conducted on an Illumina NovaSeq 6000 platform (Illumina Inc., San Diego, CA, USA) with 2 × 150 bp paired-end reads, following library preparation using a TruSeq Nano DNA library prep kit (Illumina, Inc., San Diego, CA, USA). Sequencing was performed by a commercial service provider (DNA Link, Inc., Seoul, Republic of Korea). Gene prediction of coding sequences (CDSs), rRNAs, and tRNAs was performed using Prokka v1.13. Species identity was confirmed based on average nucleotide identity (ANI) analysis using JSpecies v1.2.1 by comparing the genome to multiple reference strains. Antimicrobial resistance genes were identified using the ResFinder database (version 4.7.2).

### 2.10. Safety

#### 2.10.1. Cytotoxicity to Intestinal Epithelial Cells

The cytotoxicity of the strain was evaluated using the Quanti-LDH PLUS cytotoxicity assay kit (Biomax, Seoul, Republic of Korea) according to the manufacturer’s instructions. Briefly, HT-29 cells were seeded in a 96-well plate at a density of 2.5 × 10^4^ cells/well. After 24 h, the cells were treated with the WC of *L. fermentum* MG4717 (10^6^–10^8^ CFU/mL) for 24 h. Absorbance was measured at 490 nm using a microplate reader.

#### 2.10.2. Hemolytic Activity

*L. fermentum* MG4717 was streaked onto tryptic soy agar (Difco) supplemented with 5% sheep blood (MBcell, Seoul, Republic of Korea) and incubated at 37 °C for 48 h. Then, hemolytic activity was assessed as follows. The formation of a greenish zone, a transparent zone, and the absence of a zone around the bacterial colonies indicated α-hemolysis, β-hemolysis, and γ-hemolysis.

#### 2.10.3. Bile Salt Hydrolase (BSH) Assay

*L. fermentum* MG4717 was inoculated onto MRS agar containing 0.5% sodium glycodeoxycholate and 0.5% taurodeoxycholate. The BSH activity was determined based on the formation of precipitate zones around the colonies, indicating bile salt hydrolysis.

#### 2.10.4. Antibiotic Susceptibility Test

The antibiotic resistance of *L. fermentum* MG4717 was determined using the minimum inhibitory concentration (MIC) method. *L. fermentum* MG4717 was cultured in MRS agar at 37 °C for 48 h. The colonies were harvested, resuspended in PBS, and adjusted to a 0.5 McFarland standard. After, the cell suspensions were inoculated onto Mueller–Hinton agar and LAB susceptibility test medium (90% iso-Sensitest broth, 10% MRS broth, and 1.7% agar). MIC test strips (Liofilchem, Inc., Roseto degli Abruzzi, Italy) were placed on the agar surface and incubated at 37 °C for 48 h. Antibiotic susceptibilities were determined according to the guidelines of the European Food Safety Authority (EFSA) [21].

### 2.11. Probiotic Properties

#### 2.11.1. Scanning Electron Microscopy (SEM)

The morphological characteristics of *L. fermentum* MG4717 were confirmed using field-emission scanning electron emission SEM (FE-SEM SU5000; Hitachi, Tokyo, Japan). The SEM images were observed at 10,000 × magnification with an accelerating voltage of 3.0 kV.

#### 2.11.2. Hydrogen Peroxide (H_2_O_2_) Production

H_2_O_2_ production was performed as previously described [1]. To evaluate H_2_O_2_ production, *L. fermentum* MG4717 was cultured on MRS agar supplemented with 3,3′,5,5′-tetramethylbenzidine. The plates were then incubated under ambient air for 2 h to facilitate the detection of H_2_O_2_ produced by *L. fermentum* MG4717.

#### 2.11.3. Adhesion Ability on HT-29 Cells

Adhesion to HT-29 cells was performed as previously reported with some modifications [22]. Briefly, HT-29 cells were seeded at 1.5 × 10^5^ cells/well in 12-well plates until a cellular monolayer formed. The cells were then incubated with *L. fermentum* MG4717 (1 × 10^8^ CFU/mL) for 2 h. After incubation, non-adherent bacteria were removed by washing the wells thrice with PBS, and the cells were lysed in 1 mL of PBS. The adhesion rate (%) was calculated by comparing the number of adherent cells to the initial number of viable cells using the following formula: log (adherent counts, CFU/mL)/log (initial counts, CFU/mL) × 100.

#### 2.11.4. Survival in Simulated Gastrointestinal Tract (GIT) Conditions

Artificial gastric and intestinal fluid samples were used to evaluate the survival of *L. fermentum* MG4717 in the human gastrointestinal tract. The GIT conditions were investigated following a modified version of a previously described method [23]. To simulate GIT conditions, *L. fermentum* MG4717 (1 × 10^8^ CFU/mL) was treated in PBS (pH 2.5) containing 0.3% pepsin (Sigma-Aldrich, St. Louis, MO, USA) and incubated for 2 h at 37 °C. Then, *L. fermentum* MG4717 was treated in PBS (pH 7.4) supplemented with 1% pancreatin and 1% bile salt (Sigma-Aldrich) and incubated for 2.5 h at 37 °C. After incubation, the survival rate of *L. fermentum* MG4717 was determined by counting live colonies on the MRS agar. The initial colony count was measured before treatment, and the colony counts were measured after each treatment. The survival rate was calculated using the following formula: Survival rate (%) = log (survival counts, CFU/mL)/log (initial counts, CFU/mL) × 100.

### 2.12. Statistical Analysis

Data are presented as the mean ± standard error of the mean. The normality of distribution was confirmed using the Shapiro–Wilk test. Statistical analysis was conducted using Student’s *t*-test with Prism 10.4.1 software (GraphPad Software, San Diego, CA, USA). Statistical significance was set at *p* < 0.05.

## 3. Results

### 3.1. L. fermentum Strains Derived from the Oral Cavity Inhibit the Growth and Biofilm Formation of Oral Pathogens

Growth inhibition of *S. mutans*, *A. actinomycetemcomitans*, and *P. gingivalis* by the CFS of *L. fermentum* strains was confirmed (Figure 1A). The growth of *S. mutans* was significantly inhibited by all *L. fermentum* strains. The growth rate of *A. actinomycetemcomitans* was significantly reduced by *L. fermentum* strains MG4697, MG4712, MG4717, and MG4737. Additionally, the growth rate of *P. gingivalis* was significantly inhibited by all *L. fermentum* strains except for MG4737. Based on the growth inhibition results against oral pathogens, *L. fermentum* strains MG4684, MG4712, and MG4717 were selected to confirm the inhibition rate of biofilm formation. The selected *L. fermentum* strains were confirmed to inhibit the biofilm formation of oral pathogens (Figure 1B). *L. fermentum* MG4717 significantly inhibited the biofilm formation against *S. mutans*. Among all strains, *L. fermentum* MG4717 exhibited the highest inhibition of *A. actinomycetemcomitans*, with all strains considerably reducing biofilm formation by more than 50%. Biofilm formation by *P. gingivalis* was decreased by *L. fermentum* MG4717, although this decrease was not statistically significant. *L. fermentum* MG4717 showed the most potent inhibitory effect on periodontopathogen growth and biofilm formation. Therefore, *L. fermentum* MG4717 was selected for subsequent experiments.

### 3.2. Inhibitory Effect of L. fermentum MG4717 on the Growth of P. gingivalis and F. nucleatum, and Downregulation of mgl mRNA Expression of P. gingivalis

As shown in Figure 2A, a clear inhibition zone formed around the paper disks containing the culture medium of *L. fermentum* MG4717, indicating that *L. fermentum* MG4717 has a direct inhibitory effect on the growth of *P. gingivalis*. The culture supernatant of *L. fermentum* MG4717 showed no inhibition. The mRNA levels of *mgl* in *P. gingivalis* treated with *L. fermentum* MG4717 are shown in Figure 2B. *L. fermentum* MG4717 significantly inhibited *mgl* expression against *P. gingivalis* compared with that in the untreated control (*p <* 0.001).

In addition, as shown in Figure 2C,D, *L. fermentum* MG4717 significantly inhibited the growth of *F. nucleatum* ssp. *nucleatum* and *F. nucleatum* ssp. *animalis*. The growth rate was reduced to approximately 55% and 65%, respectively, compared with the untreated control (*p* < 0.001).

### 3.3. Adhesion Ability of L. fermentum MG4717 on Oral Epithelial Cells

The ability of *L. fermentum* MG4717 to adhere to oral epithelial KB cells is summarized in Table 2. After 2 h of incubation with *L. fermentum* MG4717, the adhesion rate to oral epithelial cells was 84.73 ± 0.33.

### 3.4. Genome Analysis of L. fermentum MG4717

Genomic characterization of *L. fermentum* MG4717 revealed a single circular chromosome measuring 2,189,996 bp with a GC content of 51.32% (Figure 3). The annotated genome comprised 2146 predicted CDSs along with 15 rRNA genes, comprising five copies each of the 5S, 16S, and 23S rRNA genes and 61 transfer RNA genes. A DNA plot was generated to depict genomic organization, functional annotation, and overall chromosomal structure.

ANI analysis confirmed the taxonomic assignment of MG4717 as *L. fermentum*, with 98.73% similarity to the type strain *L. fermentum* DSM20052, as determined using JSpecies v1.2.1 (Table 3). Furthermore, *L. fermentum* MG4717 lacked acquired antibiotic resistance genes in its genome, as determined using the ResFinder database.

### 3.5. Morphology and Safety of L. fermentum MG4717

The morphological characteristics of *L. fermentum* MG4717, which showed a distinct rod-shaped structure, were examined using SEM. *L. fermentum* MG4717 formed a rod-shaped foam (Figure 4A). To investigate the safety of *L. fermentum* MG4717, its cytotoxicity in HT-29 cells, hemolysis, and BSH activity were evaluated. *L. fermentum* MG4717 (10^6^–10^8^ CFU/mL) showed no cytotoxicity (≥100%) in HT-29 cells (Figure 4B). Moreover, *L. fermentum* MG4717 has no hemolytic activity (γ-hemolysis) and no BSH activity (Figure 4C,D).

The antibiotic susceptibility of *L. fermentum* MG4717 was evaluated according to EFSA guidelines. The MICs of the tested antibiotics were notably below the microbiological cut-off values, indicating the absence of acquired antimicrobial resistance (Table 4). Therefore, the safety of *L. fermentum* MG4717 was confirmed to be suitable for use as a probiotic.

### 3.6. GIT Stability and Adhesion to HT-29 Cells of the L. fermentum MG4717

We determined the stability of *L. fermentum* MG4717 in a simulated GIT (Table 5). The initial count was 8.21 ± 0.01 Log CFU/mL. After exposure to the simulated gastrointestinal tract, the count was confirmed at 7.40 ± 0.04 Log CFU/mL. In the simulated gastric fluid, *L. fermentum* MG4717 maintained a viable count of 7.20 ± 0.04 Log CFU/mL. These results demonstrated that *L. fermentum* MG4717 exhibited considerable resistance to harsh GIT environments, retaining over 87% viability throughout the simulation. Additionally, we confirmed the ability of *L. fermentum* MG4717 to adhere to HT-29 cells. The initial counts were 8.74 ± 0.03 Log CFU/mL, and the number of adherent cells was 6.55 ± 0.05 Log CFU/mL, resulting in an adhesion rate of 89.0 ± 1.77%.

## 4. Discussion

The oral cavity is the second most microbially diverse organ in the human body and harbors bacteria, fungi, and viruses that collectively influence oral and systemic health [24]. Imbalances in the oral microbiota have been closely linked to diseases such as dental caries and periodontitis, both of which are driven mainly by pathogenic biofilm formation on tooth surfaces [25]. In particular, *P. gingivalis* and *A. actinomycetemcomitans* are recognized as periodontopathogens, with virulence factors, such as gingipains, playing major roles in tissue destruction and immune modulation [26]. *P. gingivalis*, a key periodontopathogenic bacterium, contributes to periodontal disease through multiple virulence factors, including gingipains and cysteine proteases that facilitate tissue destruction, immune evasion, and biofilm maturation [26,27]. *L. fermentum* MG4717 significantly inhibited the growth of *S. mutans*, *A. actinomycetemcomitans*, and *P. gingivalis* and notably reduced biofilm formation, especially by *S. mutans* and *A. actinomycetemcomitans*. Additionally, a disk diffusion assay confirmed the direct antibacterial activity of *L. fermentum* MG4717 against *P. gingivalis*. These results are consistent with those of previous studies, demonstrating that *Lactobacillus* species can exert strong antibacterial effects by producing organic acids, hydrogen peroxide, and other antimicrobial compounds [28].

H_2_O_2_-producing *Lactobacillus* strains suppress the growth of *S. mutans* by inducing oxidative stress and disrupting bacterial membranes [6]. In this study, H_2_O_2_ production by *L. fermentum* MG4717 was confirmed using 3,3′,5,5′-tetramethylbenzidine agar plates (Appendix A). H_2_O_2_ production may contribute to the inhibition of oral pathogen growth and biofilm formation. Halitosis is often associated with VSCs produced by the proteolytic activity of certain oral bacteria [29].

Recent studies have highlighted a close association between *P. gingivalis* and *mgl* expression, a gene implicated in metabolic processes that are potentially linked to pathogenicity [30]. Elevated levels of methylglyoxal have been linked to periodontal tissue destruction and VSC formation, which are the key contributors to halitosis [31]. The *mgl* expression by *P. gingivalis* is a metabolic byproduct and may also function as a virulence mechanism, enhancing its persistence in the inflammatory periodontal pocket [30]. In this study, co-cultivation with *L. fermentum* MG4717 and *P. gingivalis* resulted in a significant decrease in the expression of the *mgl* mRNA levels, which may have inhibited the pathogenicity of *P. gingivalis* through direct or indirect interactions. Considering that *mgl*-associated metabolites involved in the biosynthesis of VSCs have been implicated in halitosis, the downregulation of *mgl* expression indicates that *L. fermentum* MG4717 may have an indirect inhibitory effect on halitosis. *F. nucleatum* has been reported to play a role in biofilm community through its ability to interact with a wide range of microbial species and mediate coaggregation [32]. Inhibition of *F. nucleatum*, which is responsible for periodontitis and halitosis, is particularly important given its established role in biofilm formation, tissue invasion, and volatile sulfur compound (VSC) production [33]. In this study, *L. fermentum* MG4717 exhibited a significant inhibitory effect on the growth of both *F. nucleatum* ssp. *nucleatum* and ssp. *animalis*, further supporting its broad-spectrum antimicrobial activity against major oral pathogens.

Oral epithelial cells are the first natural barrier of the periodontal tissue, and the penetration of periodontal pathogens into these cells is a crucial step in the pathogenesis of periodontal disease [34]. The adhesion to epithelial cells is a critical feature of effective oral probiotics, allowing them to compete with pathogens for colonization sites and enhancing local immune responses [1,35]. The human tongue epithelium, which is rich in papillae, facilitates the uptake of small molecules such as postbiotic metabolites and mediates interactions with host tissues [36]. Probiotic strains capable of adhering to oral epithelial cells are generally more likely to survive in the oral cavity and exert beneficial effects. *Weissella cibaria* oraCMU has demonstrated strong adhesion to oral epithelial cells and competitive exclusion of periodontal pathogens such as *P. gingivalis* and *F. nucleatum*, leading to clinically validated improvements in oral health [37]. According to previous studies, a bacterial adhesion count of more than 1.5 per KB indicates a very strong adhesion capability [38]. *L. fermentum* MG4717 exhibited a high adhesion rate to KB cells, suggesting its potential to persist in the oral cavity and maintain microbial balance.

WGS is a genomic tool that sequences the entire genome of microorganisms and compares it to existing genetic databases to provide insights into their functional characteristics, including potential virulence and metabolic capabilities [39]. ANI analysis is a valuable method for identifying bacterial species and understanding the structure of bacterial populations. It can group genomes into the same or different species using a 95% ANI cut-off point [40]. In our results, MG4717 was confirmed as *L. fermentum*, with 98.73% similarity to the type strain *L. fermentum* DSM20052. Safety is a prerequisite of any candidate probiotic strain [41]. Probiotic strains exhibiting high levels of resistance may pose safety concerns, such as the risk of horizontal transfer of antimicrobial resistance genes to co-residing pathogens in the microbiota [42]. As WGS has become more widely used, it is increasingly used to predict antimicrobial resistance phenotypes and identify resistance-related genes [43]. Using the ResFinder database, *L. fermentum* MG4717 confirmed the absence of transferable resistance genes. Additionally, the antibiotic susceptibility profile of *L. fermentum* MG4717 showed that the MIC values for the eight tested antibiotics remained below the EFSA microbiological cut-off values, indicating no evidence of acquired antimicrobial resistance genes. According to the FAO/WHO and EFSA guidelines, the absence of hemolysis and BSH activity is crucial for minimizing risks such as erythrocyte damage or excessive cholesterol deconjugation in vivo [1,2]. *L. fermentum* MG4717 satisfied the major safety criteria, showing no cytotoxicity in HT-29 cells or hemolytic BSH activity.

Probiotic strains must remain viable within the product during consumption and endure the harsh conditions of the GIT, including exposure to gastric acid, bile salts, and digestive enzymes such as pepsin, lipase, and pancreatin, to exert beneficial effects on the host [44]. Probiotics should also survive transit through the upper GI tract and subsequently colonize the intestinal epithelium [45]. *L. fermentum* MG4717 demonstrated high viability under simulated gastrointestinal conditions, with high tolerance to acidic and bile environments, and showed >74% adhesion to intestinal HT-29 cells. Probiotics with high adhesion abilities can competitively attach to the intestinal mucosa, thereby inhibiting pathogen colonization [46]. Thus, *L. fermentum* MG4717, which exhibits no hemolytic activity and lacks BSH activity, meets the key safety criteria and demonstrates high stability under both simulated GIT conditions and in intestinal epithelial cells, supporting its potential as a probiotic.

## 5. Conclusions

In this study, *L. fermentum* MG4717 demonstrated antimicrobial effects against *S. mutans*, *A. actinomycetemcomitans*, *P. gingivalis*, and *F. nucleatum,* with the ability to inhibit biofilm formation. *L. fermentum* MG4717 inhibited the expression of the *mgl* mRNA levels, which are involved in gingivitis and the pathogenicity of *P. gingivalis.* It also strongly adhered to oral epithelial cells, indicating its potential to colonize the oral cavity. In addition, *L. fermentum* MG4717 met the essential criteria for safety and probiotics. These results support the potential use of *L. fermentum* MG4717 as a safe and effective probiotic for maintaining oral health and preventing oral diseases. However, this study confirmed the potential of *L. fermentum* MG4717 as an effective probiotic for oral diseases and halitosis caused by oral microbiota imbalance in the preclinical stage. Therefore, additional in vivo and clinical studies are required to evaluate the anti-inflammatory activity in periodontal disease and the action in the oral environment, which will be supplemented in future studies.

## Figures and Tables

**Figure 1 microorganisms-13-01600-f001:**
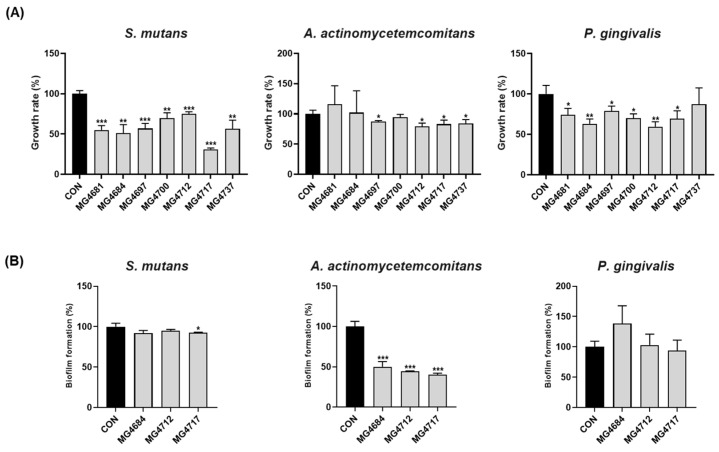
Antimicrobial activity of *L. fermentum* strains growth rate (**A**) and biofilm formation (**B**) against *S. mutans*, *A. actinomycetemcomitans*, and *P. gingivalis*. Data are presented as mean ± standard error of the mean (*n* = 3) and were analyzed using Student’s *t*-test, * *p* < 0.05, ** *p* < 0.01, and *** *p* < 0.001 vs. CON.

**Figure 2 microorganisms-13-01600-f002:**
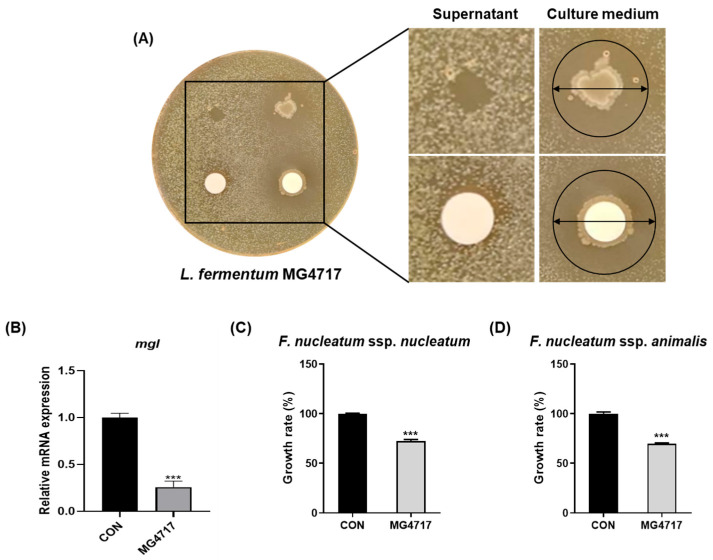
Inhibitory effect of *L. fermentum* MG4717 on *P. gingivalis* growth (**A**), the expression of the *mgl* levels (**B**), *F. nucleatum* ssp. *nucleatum* (**C**), and *F. nucleatum* ssp. *animalis* (**D**). Growth inhibition of *L. fermentum* MG4717 against *P. gingivalis* was evaluated using the disk diffusion assay. The formation of a clear inhibition zone around the disk containing *L. fermentum* MG4717, indicates by arrows, demonstrates antibacterial activity. Data are presented as mean ± standard error of the mean (*n* = 3) and were analyzed using Student’s *t*-test, *** *p* < 0.001 vs. CON.

**Figure 3 microorganisms-13-01600-f003:**
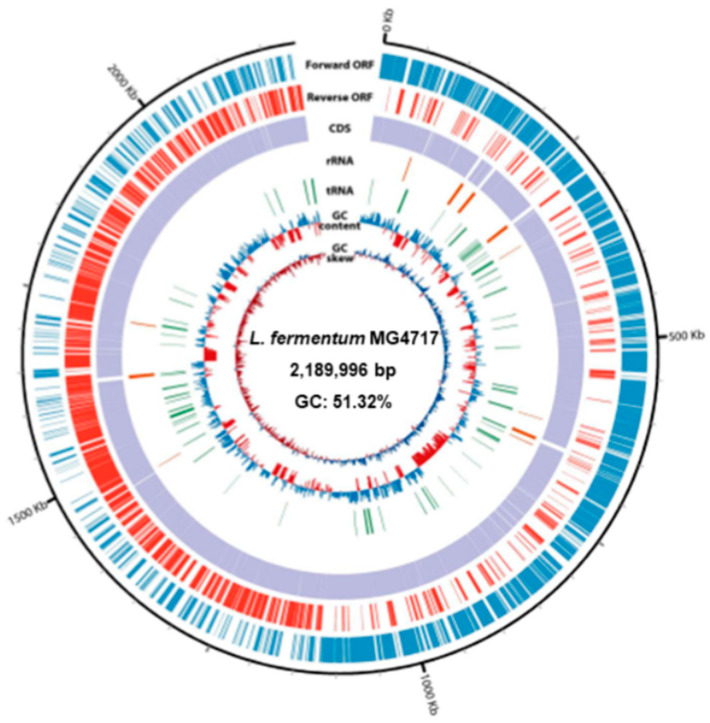
Circular genome map of *L. fermentum* MG4717. From the outer ring to the center, the plot illustrates the following genomic elements: coding sequences (CDSs) on the forward strand, CDSs on the reverse strand, tRNA loci, rRNA operons, GC content variation, and GC skew distribution.

**Figure 4 microorganisms-13-01600-f004:**
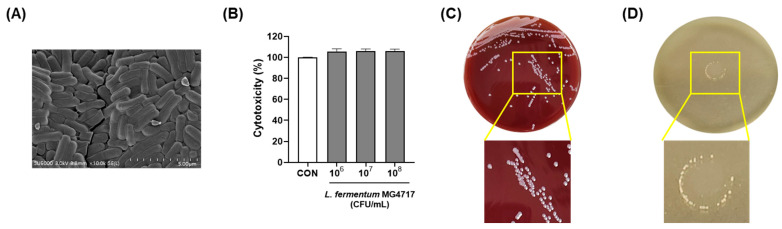
Morphology and safety of *L. fermentum* MG4717. SEM image (**A**), cytotoxicity (**B**), hemolytic activity (**C**), and BSH activity (**D**). The scale bar of the SEM image represents 5 μm.

**Table 1 microorganisms-13-01600-t001:** Accession numbers and origins of the *L. fermentum* strains used in this study.

Bacteria	Strain	NCBI Accession Number	Origin
*Limosilactobacillus fermentum*	MG4681	OP035515	Human (oral)
	MG4684	OP035518	
	MG4697	OP077100	
	MG4700	OP077103	
	MG4712	OP077115	
	MG4717	OP035525	
	MG4737	OP035543	

**Table 2 microorganisms-13-01600-t002:** Adhesion ability of *L. fermentum* MG4717 on oral epithelial KB cells.

Strain	No. of Adhered Strains Per One Epithelial Cell	Adhesion Rate (%)
*L. fermentum* MG4717	10.56 ± 0.56	84.73 ± 0.33

All data are presented as the mean ± standard error of the mean (*n* = 3).

**Table 3 microorganisms-13-01600-t003:** ANI values between the genomes of type strains belong to the genus *Limosilactobacillus*

Species	Strain	ANI (%)
*Limosilactobacillus fermentum*	DSM20052	98.73
*Limosilactobacillus reuteri* subsp. *rodentium*	100-23	80.57
*Limosilactobacillus reuteri* subsp. *reuteri*	JCM1112	79.90
*Limosilactobacillus reuteri* subsp. *kinnaridis*	AP3	77.75

**Table 4 microorganisms-13-01600-t004:** Antibiotic susceptibility of *L. fermentum* MG4717.

Antibiotics	MIC (µL/mL)	Cut-Off Value (µL/mL) ^1^
Ampicillin	0.19	2
Gentamicin	0.25	16
Kanamycin	6	64
Streptomycin	6	64
Tetracycline	2	8
Chloramphenicol	3	4
Erythromycin	0.94	1
Vancomycin	-	n.r.
Clindamycin	0.32	4

^1^ Microbiological cut-off values for antibiotics against *L. fermentum* as provided by the EFSA guidelines (2023). n.r.; not required.

**Table 5 microorganisms-13-01600-t005:** The survival rate of *L. fermentum* MG4717 in simulated gastrointestinal fluid and the adhesion rate to HT-29 cells.

Experiments		*L. fermentum* MG4717
GIT (Log CFU/mL)	Initial counts	8.21 ± 0.01
	Gastric fluid	7.40 ± 0.04
	Gastrointestinal tract	7.20 ± 0.04
Adhesion (Log CFU/mL)	Initial	8.74 ± 0.03
	Adherent	6.55 ± 0.05
Adhesion rate (%)		89.0 ± 1.77

All data are presented as the mean ± standard error of the mean (*n* = 3).

## Data Availability

The original contributions presented in this study are included in the article and Appendix A. Further inquiries can be directed to the corresponding author.

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
