# Peer review of "The Effect of Limosilactobacillus fermentum MG4717 on Oral Health and Biosafety"

_microorganisms, 2025, doi:10.3390/microorganisms13071600_

Round 1
Reviewer 1 Report
Comments and Suggestions for Authors
Please find my suggestions as a separate word document
Lines 8 and 9
Abstract: Oral diseases such as periodontitis, dental caries, and halitosis are closely asso- 8
ciated with dysbiosis of the oral microbiota and continue to pose significant public health 9
challenges worldwide.
Halitosis is not a oral disease, but condition associated with them. Please rewrite the sentence
Lines 30 and 31
Oral diseases encompass a comprehensive range of conditions, including caries, per- 30
iodontal disease, tooth loss, and halitosis, and are chronic diseases with a high prevalence 31
worldwide [1].
Rewrite this sentence: Only caries and periodontal disease are diseases.
Lines 36-38
Periodontitis is an inflammatory disease of the gingiva, surrounding tissues, and al- 36
veolar bone triggered by periodontitis-associated bacteria that rapidly proliferate after 37
colonizing the dental calculus on the tooth surface, leading to inflammation of the sup- 38
porting periodontal structures [3].
Although this is common knowledge, the sentence is not clear. Please rewrite it as Periodontitis is an inflammatory disease of tooth surrounding tissues i.e. gingiva and alveolar bone….
Lines 39 and 40
Periodontitis is primarily initiated and caused by Fuso- 39
bacterium nucleatum and Porphyromonas gingivalis (P. gingivalis), which are classified as or- 40
ange and red complexes [4].
When authors mention species for the first time, the complete name should stand. After that they can use abbreviation without mentioning it for the first time in parenthesis.
Lines 46-54
Halitosis is defined as an unpleasant odor originating from the mouth, caused by L- 46
cysteine and L-methionine produced by food intake [6]. P. gingivalis, a representative 47
gram-negative anaerobic bacterium, is a major pathogen that produces volatile sulfur 48
compounds (VSCs) such as hydrogen sulfide and methyl mercaptan, thereby contributing 49
to halitosis [7,8]. Streptococcus mutans (S. mutans), a gram-positive facultative anaerobic 50
bacterium, forms an initial oral biofilm community by interacting with other pathogens 51
via specific adhesion receptor mechanisms [9]. This environment may promote the growth 52
and metabolic activity of VSC-producing bacteria, indirectly contributing to halitosis [10]. 53
Moreover, the acidic microenvironment created by S. mutans can lead to dysbiosis, favor- 54
ing the proliferation of anaerobic bacteria implicated in malodor production [11].
Additional details about the S. mutans role in oral diseases may be incorporated in this section
Line 69
Commonly studied oral probiotics include S. salivarius, Limosilactobacillus reuteri, and 69
Limosilactobacillus fermentum (L. fermentum),
Provide a full name of S.salivarius
Lines 127-129
After incubation, the plates were washed thrice with distilled water 127
and dried. Then, the oral pathogens were stained with 0.1% crystal violet for 2 min, 128
Microorganisms 2025, 13, x FOR PEER REVIEW 4 of 15
washed three times with distilled water, dried, and dissolved in 95% ethanol. The absorb- 129
ance of each well was measured at 575 nm using a microplate reader (BioTek).
Why only 2 min? Isn’t 2min to short for the staining? This can affect the results also.
Lines 117- 129
2.4. Biofilm Formation 117
To evaluate the inhibitory effects of L. fermentum strains on biofilm formation by oral 118
pathogens, a crystal violet assay was performed with a few modifications [19]. Oral path- 119
ogens were diluted to a concentration of 1 × 108 CFU/mL. S. mutans (1 × 104 CFU/well) was 120
inoculated onto a 96-well plate under anaerobic conditions for 12 h. Then, 10% CFS of each 121
strain was treated and cultured for an additional 24 h. Then, 100 μ L of A. actinomycetem- 122
comitans (1 × 107 CFU/well) was inoculated onto a 96-well plate under anaerobic conditions 123
for 24 h. Additionally, 10% CFS were then treated and cultured for an additional 24 h. P. 124
gingivalis was inoculated onto a 96-well plate at a concentration of 2 × 106 CFU/well and 125
incubated for five days. Then, the plates were cultured for an additional 24 h with 10% 126
CFS from each strain. After incubation, the plates were washed thrice with distilled water 127
and dried. Then, the oral pathogens were stained with 0.1% crystal violet for 2 min, 128
Microorganisms 2025, 13, x FOR PEER REVIEW 4 of 15
washed three times with distilled water, dried, and dissolved in 95% ethanol. The absorb- 129
ance of each well was measured at 575 nm using a microplate reader (BioTek
The section needs to be rewritten to be clear how this was done. Please precise clearly time dynamic per species.
Author Response
<GENERAL COMMENTS>
Point 1: Lines 8 and 9 Abstract: Oral diseases such as periodontitis, dental caries, and halitosis are closely asso- 8 ciated with dysbiosis of the oral microbiota and continue to pose significant public health 9 challenges worldwide. Halitosis is not a oral disease, but condition associated with them. Please rewrite the sentence.
Response 1: Thanks for your comments. We changed the sentence in lines 8-10.
Point 2: Lines 30 and 31. Oral diseases encompass a comprehensive range of conditions, including caries, per- 30 iodontal disease, tooth loss, and halitosis, and are chronic diseases with a high prevalence 31 worldwide [1]. Rewrite this sentence: Only caries and periodontal disease are diseases.
Response 2: Thanks for your comments. We changed the sentence in lines 32-34.
Point 3: Lines 36-38 Periodontitis is an inflammatory disease of the gingiva, surrounding tissues, and al- 36 veolar bone triggered by periodontitis-associated bacteria that rapidly proliferate after 37 colonizing the dental calculus on the tooth surface, leading to inflammation of the sup- 38 porting periodontal structures [3]. Although this is common knowledge, the sentence is not clear. Please rewrite it as Periodontitis is an inflammatory disease of tooth surrounding tissues i.e. gingiva and alveolar bone…
Response 3: Thanks for your comments. We changed the sentence in lines 38-41.
Point 4: Lines 39 and 40 Periodontitis is primarily initiated and caused by Fuso- 39 bacterium nucleatum and Porphyromonas gingivalis (P. gingivalis), which are classified as or- 40 ange and red complexes [4]. When authors mention species for the first time, the complete name should stand. After that they can use abbreviation without mentioning it for the first time in parenthesis.
Response 4: Thanks for your comments. We have revised the sentence in line 42 to use the full species names followed by their abbreviations upon first mention.
Point 5: Lines 46-54 Halitosis is defined as an unpleasant odor originating from the mouth, caused by L- 46 cysteine and L-methionine produced by food intake [6]. P. gingivalis, a representative 47 gram-negative anaerobic bacterium, is a major pathogen that produces volatile sulfur 48 compounds (VSCs) such as hydrogen sulfide and methyl mercaptan, thereby contributing 49 to halitosis [7,8]. Streptococcus mutans (S. mutans), a gram-positive facultative anaerobic 50 bacterium, forms an initial oral biofilm community by interacting with other pathogens 51 via specific adhesion receptor mechanisms [9]. This environment may promote the growth 52 and metabolic activity of VSC-producing bacteria, indirectly contributing to halitosis [10]. 53 Moreover, the acidic microenvironment created by S. mutans can lead to dysbiosis, favor- 54 ing the proliferation of anaerobic bacteria implicated in malodor production [11]. Additional details about the S. mutans role in oral diseases may be incorporated in this section
Response 5: Thanks for your comments. We changed the sentence in lines 53-57.
Point 6: Line 69 Commonly studied oral probiotics include S. salivarius, Limosilactobacillus reuteri, and 69 Limosilactobacillus fermentum (L. fermentum), Provide a full name of S.salivarius
Response 6: Thanks for your comments. We corrected “S. salivarius” to “Streptococcus salivrius” in line 75.
Point 7: Lines 127-129 After incubation, the plates were washed thrice with distilled water 127 and dried. Then, the oral pathogens were stained with 0.1% crystal violet for 2 min, 128
Microorganisms 2025, 13, x FOR PEER REVIEW 4 of 15 washed three times with distilled water, dried, and dissolved in 95% ethanol. The absorb- 129 ance of each well was measured at 575 nm using a microplate reader (BioTek). Why only 2 min? Isn’t 2min to short for the staining? This can affect the results also.
Response 7: Thanks for your comments. We apologize for any confusion. The dyeing time was incorrectly listed as 2 min. Based on the method described above, we have corrected the correct dyeing time to 15 minutes on line 144 [1].
Point 8: Lines 117- 129 2.4. Biofilm Formation 117 To evaluate the inhibitory effects of L. fermentum strains on biofilm formation by oral 118 pathogens, a crystal violet assay was performed with a few modifications [19]. Oral path- 119 ogens were diluted to a concentration of 1 × 108 CFU/mL. S. mutans (1 × 104 CFU/well) was 120 inoculated onto a 96-well plate under anaerobic conditions for 12 h. Then, 10% CFS of each 121 strain was treated and cultured for an additional 24 h. Then, 100 μ L of A. actinomycetem- 122 comitans (1 × 107 CFU/well) was inoculated onto a 96-well plate under anaerobic conditions 123 for 24 h. Additionally, 10% CFS were then treated and cultured for an additional 24 h. P. 124 gingivalis was inoculated onto a 96-well plate at a concentration of 2 × 106 CFU/well and 125
incubated for five days. Then, the plates were cultured for an additional 24 h with 10% 126
CFS from each strain. After incubation, the plates were washed thrice with distilled water 127 and dried. Then, the oral pathogens were stained with 0.1% crystal violet for 2 min, 128
Microorganisms 2025, 13, x FOR PEER REVIEW 4 of 15 washed three times with distilled water, dried, and dissolved in 95% ethanol. The absorb- 129 ance of each well was measured at 575 nm using a microplate reader (Biotek) The section needs to be rewritten to be clear how this was done. Please precise clearly time dynamic per species.
Response 8: Thanks for your comment. We have separated each pathogen's procedures and clarified the inoculation concentration and incubation time. Lines 130-142 also include the timing of CFS treatment.
Reference
- Widyarman, A.S.; Lay, S.H.; Wendhita, I.P.; Tjakra, E.E.; Murdono, F.I.; Binartha, C.T.O. Indonesian mangosteen fruit (Garcinia mangostana L.) peel extract inhibits Streptococcus mutans and Porphyromonas gingivalis in biofilms in vitro. Contemporary Clinical Dentistry 2019, 10, 123-128.

Reviewer 2 Report
Comments and Suggestions for Authors
Dear Authors,
Thank you for submitting your manuscript entitled “The Effect of Limosilactobacillus fermentum MG4717 on Oral Health and Biosafety.” Your study presents a thorough in vitro evaluation of the antimicrobial properties, probiotic potential, and safety profile of L. fermentum MG4717, a strain isolated from the human oral cavity.
The experimental approach is methodologically solid, and the manuscript is clearly written and well-organized. However, while the data are promising, the overall novelty of the work appears incremental, as many of the methodologies and findings—particularly concerning antimicrobial assays, biofilm inhibition, adhesion, and safety—have been previously reported either in general literature or in related work by your group.
The characterization of mgl gene downregulation in P. gingivalis is a potentially interesting angle, but this observation would be more impactful if accompanied by mechanistic insights or additional functional data to explain the biological pathway involved.
Furthermore, the lack of comparative data with established oral probiotics such as L. reuteri or S. salivarius limits the contextualization of MG4717’s relative efficacy. The absence of any in vivo validation or translational framework further restricts the applicability of your conclusions. A brief discussion on delivery formats, formulation strategies, or survival in real-life oral environments would help situate the findings in a more practical context.
These limitations do not undermine the technical quality of the manuscript, but they suggest that the work may be best framed as a pilot or preclinical characterization of a promising oral probiotic strain. With some revision to clarify its novel contribution and potential applications, the manuscript could be a useful addition to the field.
Author Response
<GENERAL COMMENTS>
Point 1: The experimental approach is methodologically solid, and the manuscript is clearly written and well-organized. However, while the data are promising, the overall novelty of the work appears incremental, as many of the methodologies and findings—particularly concerning antimicrobial assays, biofilm inhibition, adhesion, and safety—have been previously reported either in general literature or in related work by your group. The characterization of mgl gene downregulation in P. gingivalis is a potentially interesting angle, but this observation would be more impactful if accompanied by mechanistic insights or additional functional data to explain the biological pathway involved.
Response 1: Thanks for your comments. To complement the functionality of MG4717 as an oral health strain in this study, we measured the antibacterial activity of MG4717 against F. nucleatum spp. nucleatum and F. nucleatum spp. animalis, the main causative strains of periodontal disease and halitosis, added the results in lines 407-414. Accordingly, we also added the methods (lines 139-142), results (lines 313-316), figure legends (lines 318-319) and discussion (lines 414-421).
Point 2: Furthermore, the lack of comparative data with established oral probiotics such as L. reuteri or S. salivarius limits the contextualization of MG4717’s relative efficacy. The absence of any in vivo validation or translational framework further restricts the applicability of your conclusions. A brief discussion on delivery formats, formulation strategies, or survival in real-life oral environments would help situate the findings in a more practical context. These limitations do not undermine the technical quality of the manuscript, but they suggest that the work may be best framed as a pilot or preclinical characterization of a promising oral probiotic strain. With some revision to clarify its novel contribution and potential applications, the manuscript could be a useful addition to the field.
Response 2: Thanks for your comments. We agree that comparative analysis with well-established oral probiotics such as L. reuteri or S. salivarius could provide valuable context. In our previous study, L. reuteri and L. fermentum, which were isolated from the oral cavity of humans, showed the most potent activity and stability against oral pathogens. A research article on L. reuteri has already been published, and the present study aims to report the findings related to the L. fermentum strain. In addition, both strains are currently being evaluated in vivo studies. Furthermore, we plan to conduct additional studies to evaluate the efficacy and safety of L. fermentum MG4717 in functional food applications targeting oral health.
The abstract and discussion were revised to better reflect the preclinical scope and potential applications of this study.
Additionally, we revised the conclusion to clarify that this study represents a preclinical evaluation of a promising oral probiotic candidate. We also added a brief discussion on the viability of probiotics in the oral environment in lines 426–435.
